# Plant Broth- (Not Bovine-) Based Culture Media Provide the Most Compatible Vegan Nutrition for In Vitro Culturing and In Situ Probing of Plant Microbiota

Hend Elsawey [1], Sascha Patz [2], Rahma A. Nemr [1], Mohamed S. Sarhan [1], Mervat A. Hamza [1], Hanan H. Youssef [1], Mohamed R. Abdelfadeel [1], Hassan-Sibroe A. Daanaa [3], Mahmoud El-Tahan [4], Mohamed Abbas [5], Mohamed Fayez [1], Katja Witzel [6], Silke Ruppel [6] and Nabil A. Hegazi [1,*]

1  ESRU, Department of Microbiology, Faculty of Agriculture, Cairo University, Giza 12613, Egypt; hendelsawey@gmail.com (H.E.); rahma.ahmed546@yahoo.co.uk (R.A.N.); m.sabrysarhan@gmail.com (M.S.S.); mervathamza66@gmail.com (M.A.H.); hananyoussef16@gmail.com (H.H.Y.); mohamed.ra.farag@std.agr.cu.edu.eg (M.R.A.); mfayezgiza@yahoo.co.uk (M.F.)
2  Algorithms in Bioinformatics, Center for Bioinformatics, University of Tübingen, 72076 Tübingen, Germany; sascha.patz@uni-tuebingen.de
3  Department of Genetics, School of Life Science, The Graduate University for Advanced Studies (SOKENDAI), Shizuoka 411-8540, Japan; hsdaanaa@nig.ac.jp
4  Regional Center for Food and Feed (RCFF), Agricultural Research Center (ARC), Giza 12613, Egypt; eltahanmh@gmail.com
5  Department of Microbiology, Faculty of Agriculture & Natural Resources, Aswan University, Aswan 81528, Egypt; mtabbas67@gmail.com
6  Department of Plant Microbe Systems, Leibniz Institute of Vegetable and Ornamental Crops, 14979 Großbeeren, Germany; Witzel@igzev.de (K.W.); ruppel@igzev.de (S.R.)
*  Correspondence: hegazinabil8@gmail.com; Tel./Fax: +20-2-35728483

**Abstract:** Plant microbiota support the diversity and productivity of plants. Thus, cultivation-dependent approaches are indispensable for in vitro manipulation of hub taxa. Despite recent advances in high-throughput methods, cultivability is lagging behind other environmental microbiomes, notably the human microbiome. As a plant-based culturing strategy, we developed culture media based on a broth of cooked aqueous mixtures of host plants. This improved the in vitro growth of representative isolates of plant microbiota and extended the in situ recovery of plant microbiota. With clover, 16S rRNA gene sequencing of representative isolates confirmed the predominance of *Firmicutes*, *Alphaproteobacteria* and *Gammaproteobacteria,* and less frequently *Bacteroidetes* and *Actinobacteria*. Whereas bovine-based culture media (modified R2A) confined the diversity to *Firmicutes*, the plant broth-based culture media revealed a wider scope of endophytes beyond rhizobia, i.e., multiple genera such as *Chryseobacterium*, *Cronobacter*, *Kosakonia*, *Tsukamurella*, and a potentially/presumptive novel species. Matrix-assisted laser desorption/ionization time-of-flight (MADI-TOF) analysis clustered isolates according to their plant niches, the endo-phyllosphere/endo-rhizosphere. We recommend the plant broth for simplicity, reproducibility and perdurable storage, supporting future culturomics applications, good laboratory practice (GLP) and good manufacturing practice (GMP). The strategy creates an "in-situ-similis" vegan nutritional matrix to analyze microbial diversity and reveal novel microbial resources pertinent to biotechnological and environmental applications.

**Keywords:** plant microbiota; cultivation-dependent of plant microbiota; plant broth-based culture media; "in-situ-similis" culturing strategy; vegan nutrition; clover bacterial endophytes

## 1. Introduction

A broad range of biomes drive diversity, evolution and the health of our changing planet. Studying these biomes is critical to understanding the complex interactions that take place between living organisms and how the ecosystem shapes these interactions. Knowing that almost every habitat or even an organism hosts a diverse network of microorganisms, its "microbiome", such knowledge could transform our understanding of the natural world and mediate several innovations in agriculture, energy, health, the environment, and more [1]. Therefore, the Earth Microbiome Project (EMP) has been founded as a cross-discipline effort among international institutes and researchers, microbial ecologists, geneticists, microbiologists, physicists, computer scientists, mathematicians and ecosystem modelers. The EMP aims to provide the most comprehensive global assessment of microbial life [2].

Microbial communities are the most abundant members of a biome in any ecosystem, and mediate complex interactions that contribute to biogeochemical cycling. While some microbes may be free-living in the environment, others exist ubiquitously associated with higher organisms, where they may cooperate with the host and contribute to host development and health [3]. In plants, although exploring microbial diversity and unraveling their functions is crucial for supporting sustainable agricultural practices [4], the plant microbiome has received relatively little attention compared to studies on human/animal microbiomes.

The distribution of microbes within a host may be shaped by host anatomy and metabolism, which may have implications for microbe functioning [5]. Indeed, plant roots, stems and leaves are occupied by several different microbial communities that may have overlapping composition and function [6]. As a further classification, endophytic bacteria comprise microbes in the inner plant tissue, whereas epiphytes occur on plant surfaces. Distinctions are also made according to surface exposure; rhizosphere bacteria occupy the root system (below the soil surface), while phyllosphere bacteria are found in the stem and leaf (above the soil surface) [7]. These complex schemes of partitioning underscore the dynamism of microbe occupancy in the host. From a functional perspective, rhizospheric and phyllospeheric microorganisms possess several functions that include facilitating nutrient uptake, and tolerance to biotic and abiotic stresses [8,9].

Advances in both culture-dependent and culture-independent methods to study microbes have enabled progression of plant microbiome studies. Since the pioneering work of Louis Pasteur, Robert Koch, and others, early culture-dependent techniques relied on the historical and traditional culture media based on meat broth, "Nutrient Broth", "Bouillon", "Nährflüssigkeit", with or without further supplements of animal/bovine origin [10–12]. Such nutrient-rich culture media long represented the basis for isolating and culturing microbes in various environments. Over time, unprecedented efforts were exerted to develop myriad formulas of standard chemically-defined and artificial culture media to fundamentally improve cultivability of microorganisms in various environments. However, the nutrient composition of such culture media was biased towards fast-growing bacteria and likely did not provide an accurate reflection of how the microbes behave *in planta* [13].

Since such culturing strategies for plant microbes have obscured our knowledge about plant microbiome diversity, more recently, revolutionary techniques have been developed involving both diversifying culture conditions and mimicking the host conditions of microbes [14,15]. These techniques attempt to employ in vitro culturing conditions that reflect the in vivo environment (host) of the microbiota. Hence, reconciling in vitro and in vivo conditions is essential to broaden the outlook of the plant microbiome [15]. Indeed, the in situ cultivation of plant microbes in their natural environments using membranes and chambers (e.g., soil substrate membrane [16], diffusion chamber [17] and hollow-fiber membrane chamber device [18]) has allowed for the isolation of several novel members of

the plant microbiome. Importantly, a more recent and promising approach has arisen through the use of plant materials or extracts as basic supplements for culture media [19–26]. Furthermore, dehydrated powders of mixed vegetables and pulses proved to support pre- and culture cultivation of bacterial species of probiotic actions, e.g., *Lactobacillus* sp. and *Bifidobacteria* sp. [27].

Initial molecular biological methods used for studying microbes emerged from the polymerase chain reaction (PCR), restriction fragment length polymorphism (RFLP) and denaturing gradient gel electrophoresis (DGGE) techniques, among others. This allowed analyses employing primers to distinguish microbes based on their 16S rRNA genes. While this approach extended the range of microbes being observed, limitations in 16S amplification caused a bias towards over-represented microbes, in other words, restricting our view to only microbes with 16S rRNA sequences complementary to the primer sequences used [28].

More recently, major advances in proteome classification using matrix-assisted laser desorption/ionization time-of-flight mass spectrometry (MALDI-TOF/MS) and DNA/RNA sequencing techniques have enabled the study and discovery of microbes via protein profiles and genome-sequence based inferences (metagenomics, proteomics, transcriptomics, etc.) [29]. However, to yield a comprehensive view of novel microbes from sequence data, and study their impact on plant development and health, a combination of both culture-dependent and -independent techniques is essential [30,31]. Nevertheless, modern techniques reveal several limitations, presenting a challenge to depict entire microbiomes accurately [32].

Our previous studies supported the idea of mimicking/simulating the natural environment of plant-associated bacteria using plant-only-based culture media prepared from plant juices, slurries, saps, and dehydrated powders [19,23]. More recently, we identified several novel bacteria and successfully cultured previously uncultured bacteria by combining plant-only-based culture media and culture-independent techniques [22,25,26,33]. To foster such a concept of "in-situ similis" culturing strategy, here we advocate the plant broth per se for simplicity, ease of preparation, reproducibility, and perdurable storage, to support future culturomic studies as well as good laboratory practice (GLP) and good manufacturing practice (GMP). We tested homologous and heterologous plant broth of clover and wheat, as culture media for in vitro culturing and in situ recovery of endophytic microbiota from two plant compartments, phyllospheres and rhizospheres. After monitoring the colony forming units (CFUs) to assess the potential of the plant-broth culture media, we explored the diversity of plant microbiota by applying 16S rRNA gene sequencing and MALDI-TOF/MS to representative isolates. These in vitro cultivated endophytic bacterial communities were compared to those recovered by standard meat/bovine-based culture media, nutrient agar and modified R2A.

## 2. Materials and Methods

### 2.1. Tested Plant Materials

The tested host plants were Berseem Clover (*Trifolium alexandrinum* L.) and wheat (*Triticum aestivum* L.). Plants were grown in open fields at the experimental station of the Faculty of Agriculture, Cairo University, Giza, Egypt (30.0131° N, 31.2089° E). Representative samples of shoots and roots of both plants about to flower were collected in plastic bags; nodules were common on roots of clover plants (Figure S1). The samples were brought to the laboratory and kept in the refrigerator prior to microbiological analyses on the same day.

### 2.2. Plant Broth (PB)

Coarse-chopped plant shoots of clover and wheat were washed and soaked in 10 L-Erlenmeyer flasks with tap water (1:2, *w/v*). After heat-extraction in an autoclave (121 °C for 20 min), the mixture was pressed and filtered through a cotton cloth to obtain a clear plant broth. Aliquots of the plant broth were stored at −20 °C until use (Figure 1).

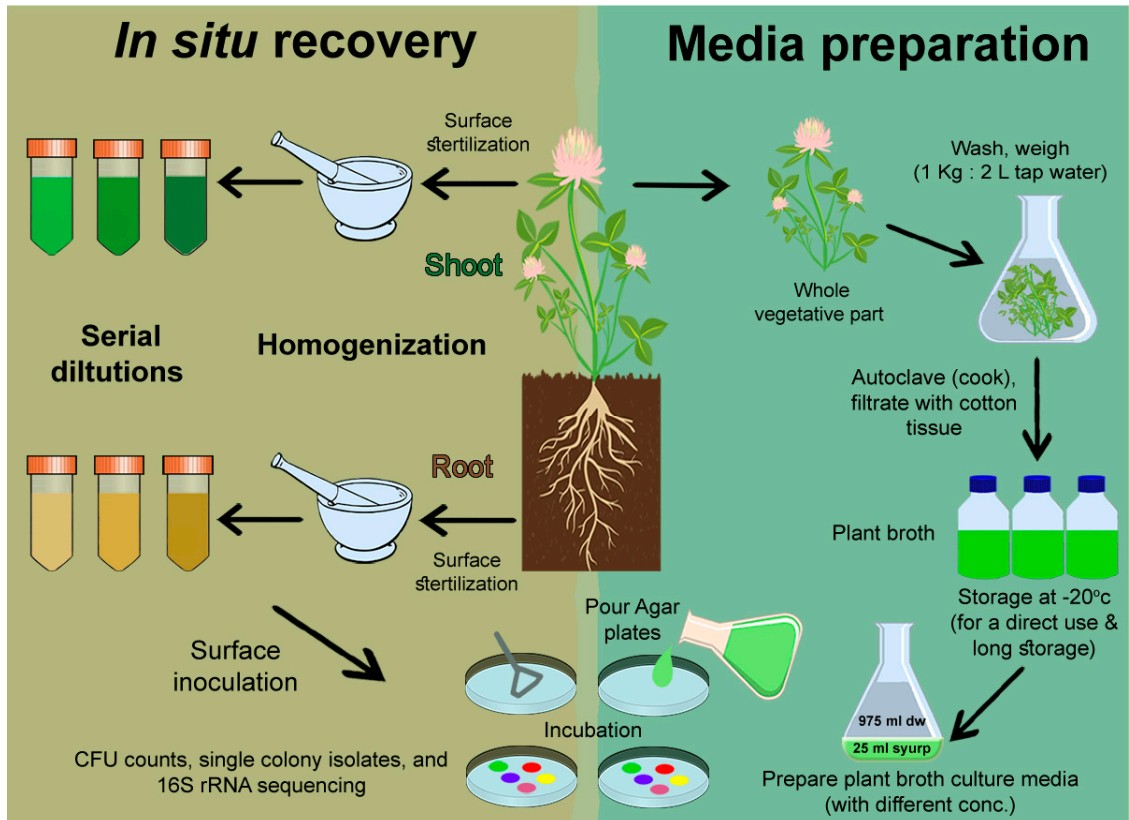

**Figure 1.** A workflow of in situ recovery and analysis of bacterial communities residing in the endo-rhizosphere and endo-phyllosphere of tested plants and plant broth culture media preparation.

## 2.3. Culture Media

Plant-Only-Based Culture Media

**Plant broth-based agar:** The plant broth culture media were prepared by the addition of different volumes (*v/v*) of the prepared plant broth to distilled water (25, 50 mL L$^{-1}$). Agar culture media were prepared by the addition of agar (2% *w/v*), then autoclaved at 121 °C for 20 min. The pH of the resulting solutions was 6.0–6.8 and kept as such without adjustment.

**Plant powder teabags culture media:** The plant powder teabags were prepared according to Sarhan [22], with concentration of 0.5 g dehydrated plant powder L$^{-1}$. Agar culture media were prepared by adding agar (2% *w/v*), then autoclaved for 20 min at 121 °C.

Standard Chemically Defined and Artificial Culture Media

**R2A agar,** with a slight modification that contains (g L$^{-1}$): casein hydrolysate, 0.5; dextrose, 0.5; soluble starch, 0.5; yeast extract, 0.5; dipotassium phosphate, 0.3; sodium pyruvate, 0.3; casein peptone, 0.25; meat peptone, 0.25; magnesium sulfate, 0.024. Agar was added (2% *w/v*) and pH adjusted to 7.0 ± 0.2 [34] (https://assets.fishersci.com/TFS-Assets/LSG/manuals/IFU112543.pdf).

**Nutrient agar** contains (g L$^{-1}$): beef extract, 3.0; peptone, 5.0; glucose, 1.0; yeast extract, 0.5; agar, 15; pH, 7.0 ± 0.2 [35].

**N-deficient combined carbon sources medium (CCM)** comprised (g L$^{-1}$): glucose, 2.0; malic acid, 2.0; mannitol, 2.0; sucrose, 1.0; K$_2$HPO$_4$, 0.4; KH$_2$PO$_4$, 0.6; MgSO$_4$, 0.2; NaCl, 0.1; MnSO$_4$, 0.01; yeast extract, 0.2; KOH, 1.5; CaCl$_2$, 0.02; FeCl$_3$, 0.015; Na$_2$MoO$_4$, 0.002. In addition, CuSO$_4$, 0.08 mg; ZnSO$_4$, 0.25 mg; sodium lactate (50% *v/v*), 0.6 mL were added. Agar was added (2% *w/v*) and pH adjusted to 7.0 ± 0.2 [36].

### 2.4. In Vitro Growth of Rhizobacteria Isolates on Plant Broth-Based Culture Media

Initially, two preliminary experiments were carried out to test the suitability of the plant broth as such, and with various concentrations to support the growth of individual isolates of rhizobacteria. For this purpose, three pure strains were selected representing the major rhizobacterial phyla of *Proteobacteria* (*Klebsiella oxytoca* and *Pseudomonas putida*) and *Firmicutes (Bacillus licheniformis)*. They were obtained from the culture collection of the Environmental Studies and Research Unit (ESRU), Department of Microbiology, Faculty of Agriculture, Cairo University, Giza, Egypt.

The first experiment measured the ability of mixed cultures of rhizobacterial strains to grow on higher concentrations of plant broth. Pure bacterial strains were separately inoculated into liquid $\frac{1}{2}$ modified R2A culture medium, and incubated at 30 °C for 24 h. Resulting broth cultures were examined microscopically for growth and purity. Then, the mixture of all tested isolates was prepared by mixing equal volumes (10 mL) of the prepared 24 hr-old bacterial cultures. Aliquots of 200 μL from the resulting mixed broth culture were evenly streaked on the surface of agar plates prepared from all tested culture media. Plant broth-based culture media were prepared by using increasing volumes of wheat and clover plant broth (50, 100, 200 and 400 mL plant broth $L^{-1}$ distilled water). For comparisons, agar plates were prepared as well from diluted nutrient agar (1:10 *v/v*) and CCM (1:2 *v/v*).

Based on the results of the first experiment, pure strains were tested for growth separately on decreasing volumes/concentrations of plant broth. Aliquots of 200 μL of each of the liquid cultures of tested isolates were evenly streaked on agar plates prepared from either fresh or long-stored (3 year-old) stocks of plant broth of clover and wheat, with final concentrations of 12.5, 25, and 100 mL plant broth $L^{-1}$ distilled water. For comparisons, agar plates of $\frac{1}{2}$ modified R2A were included as a standard culture medium.

For both preliminary experiments, replicates of 5 agar plates were prepared from each treatment, incubated at 30 °C for 2–8 days and the resulting growth was examined visually and microscopically. The growth indices recorded were: 1, scant (discontinued bacterial lawn, with scattered colonies); 2–3, good (continuous bacterial lawn); and 4–5, very good (continuous and denser bacterial lawn).

### 2.5. In Situ Cultivability of Endophytes of Plant Endo-Rhizosphere and Endo-Phyllosphere

For preparation of plant endo-rhizosphere and -phyllosphere cultures, root and leaf samples of either clover or wheat were initially washed and surface sterilized according to Youssef [37] for roots and according to de Oliveira Costa [38] and Jackson [39] for leaves. Original suspensions of roots/shoots (5 g in 45 mL basal salts of CCM culture medium as a diluent, referred to as "the mother culture", were prepared. Further serial dilutions were obtained; aliquots (200 μL) of suitable dilutions were surface inoculated on prepared agar plates, with four replicates representing all of the tested culture media. Incubation took place at 25 °C for up to 14 days, and CFUs, including micro-colonies (μCFU, <1 mm diameter determined with 40× magnification), were counted throughout. Dry weights of roots/shoots were obtained by drying the original roots/shoots suspension at 70 °C for 1–2 days.

Two main experiments were carried out to cultivate the microbiota of wheat and clover.

### 2.6. In Situ Recovery and Cultivability of Wheat Endophytes on Homologous Wheat-Based Culture Media

In this experiment, the endophytic bacterial populations of both compartments, endo-rhizosphere and endo-phyllosphere of the wheat plant, were cultured on plant-based culture media prepared from wheat broth (50 mL plant broth $L^{-1}$) and wheat powder teabags (0.5 g plant powder $L^{-1}$). Both of the standard culture media $\frac{1}{2}$ modified R2A and $\frac{1}{2}$ CCM were included for comparison. CFUs developed on surface-inoculated agar plates of all tested culture media were recorded throughout the incubation periods (1–14 days).

*2.7. Cultivability of Clover Endophytes on Culture Media Prepared from Homologous (Clover) and Heterologous (Wheat) Plant Broth*

This experiment was designed to test the cultivability of clover bacterial endophytes, of the rhizosphere and phyllosphere, on different formulations of homologous clover- and heterologous wheat-based culture media (25 and 50 mL plant broth $L^{-1}$, as well as 0.5 g plant powder $L^{-1}$). Lower concentrations of plant-broth culture media (25 mL plant broth $L^{-1}$) were used to encourage recovery of fastidious bacteria and avoid the over-growth of fast-growing bacteria. The modified R2A culture medium was used for comparison. CFUs developed on surface-inoculated agar plates were monitored throughout the incubation period of 2–6 days. For 16S rRNA gene sequencing and taxonomic assignment, all discretely developed CFUs on representative single agar plates were picked and successively sub-cultured on the corresponding culture media. The agar plates were prepared from homologous and heterologous plant-broth culture media (25 mL plant broth $L^{-1}$) and ½ modified R2A (Table S1).

*2.8. DNA Extraction and 16S rRNA Gene Sequencing of Bacterial Isolates*

Bacterial genomic DNA was extracted using QIAGEN DNeasy plant mini kit (Qiagen Inc., Hilden, Germany) according to the manufacturer's instructions. The 16S rRNA gene was amplified with the forward primer "9bfm" [5'GAGTTTGATYHTGGCTCAG-3'] and reverse primer "1512R" [5'ACGGHTACCTTGTTACGACTT-3'] [22,40]; https://www.ncbi.nlm.nih.gov/pubmed/18340335). Purified PCR products were sequenced by Eurofins MWG Operon (Ebersberg, Germany). Partial 16S rRNA gene sequences (>242–1035 bp) are deposited in the GenBank database under the accession numbers KY933298, KY953182-KY953188, KY963513-KY963522, KY974372-KY974391, MG890291-MG890302 and MG928511-MG928522.

*2.9. Protein Typing of Bacterial Isolates Using Matrix-Assisted Laser Desorption/Ionization-Time of Flight (MALDI-TOF) Mass Spectrometry*

The selected bacterial isolates developed on both clover- and wheat-broth culture media were further subjected to protein biotyping using MALDI Biotyper (Bruker Daltonics GmbH, Bremen, Germany). The tested isolates represented both plant compartments, the endo-rhizosphere and endo-phyllosphere. Total protein was extracted according to manufacturer's instructions. Pure bacterial strains were grown on the corresponding plant broth culture media overnight, and three colonies of each strain were suspended in 300 μL distilled water to which 900 μL of absolute ethanol was added. After centrifugation (18,000 rpm for 2 min at room temperature), the supernatant was completely removed, and the pellets were allowed to dry for 3 min. Then, pellets were dissolved in 10 μL of 70% formic acid, and an equal volume of acetonitrile (ACN) was added and mixed gently. The mixtures were centrifuged at 18,000 rpm for 2 min at room temperature. A volume of 1 μL of the clear supernatants were spotted in duplicate onto the MALDI target plate (Bruker Daltonics, GmbH, Bremen, Germany) and air-dried at room temperature. Each spot was overlaid with 1 μL of HCCA (a-cyano-4-hydroxy cinnamic acid) matrix solution saturated with organic solvent (50% acetonitrile and 2.5% trifluoroacetic acid) and air-dried completely. MALDI-TOF measurements were carried out using an ultrafleXtreme mass spectrometer (Bruker Daltonics GmbH, Bremen, Germany) operating in linear positive mode. Dendrograms were generated using the MALDI BioTyper software, version 3.1.

*2.10. Chemical Analysis of the Dehydrated Plant Powders*

The chemical compositions and nutritional contents of the tested plants (clover and wheat) were performed by the certified Regional Center for Food and Feed (RCFF), Agricultural Research Center (ARC), Giza, Egypt, (rcff.com.eg/ISO%20Accriditation/Scope.htm). Analyses included total crude protein, total crude fiber, total ash, total carbohydrates, amino acids, as well as macro- and micro-nutrients.

*2.11. Statistical and Phylogenetic Analyses*

For CFUs counts, analysis of variance (ANOVA) was used to examine the independent effects and interactions among incubation periods, plant spheres and culture media. Data were analyzed using the R-project packages (cran.r-project.org): "agricolae" for statistical analysis and "ggplot2" for constructing boxplots.

For phylogenetic analyses, the 16S rRNA gene sequences were taxonomically assigned by comparison with those available in GenBank nucleotide database using BlastN tool (blast.ncbi.nlm.nih.gov/Blast.cgi), EZBiocloud database (ezbiocloud.net), as well as the classifier tool of the Ribosomal Database Project (RDP) (rdp.cme.msu.edu/classifer/classifier.jspp). The obtained 16S rRNA gene sequences were aligned with Clustal Omega version 1.2.4 [41] to their closely/nearest related (NR) and respective type strain (T) sequences of the GenBank database. As outgroups *Tsukamurella* sp. (*Actinobacteria*) and *Chryseobacterium* sp. strains (*Bacteroidetes*) were chosen. The alignment was trimmed with trimAl version 1.4.rev22 (-gt 0.8 -st 0.001 -cons 70) [42]. Phylogenetic trees were constructed by using the Maximum Likelihood method under the GTRCAT model, implemented in RaxML [43]. Bootstrapping was performed on 1000 replicates, and the inferred tree was saved in Newick format and visualized with iTol (itol.embl.de) [44].

## 3. Results

*3.1. In Vitro Growth of Rhizobacteria Isolates on Plant Broth-Based Culture Media*

The chemical profile of the tested clover and wheat shoots in the form of dehydrated powders indicated that both were nutritionally rich enough in respect of macro-molecules (carbohydrates, proteins), ash and fibers (Figure 2). The diverse nutritional composition of the plants tested was further expressed in the store of amino acids as well as macro- and micro-nutrients. Such multiple nutrient matrices are highly compatible with supporting profound growth of the plant endophytes present in various plant compartments of the rhizosphere and phyllosphere. This was demonstrated by higher in vitro growth indices reported for some representatives of plant endophytes, *Klebsiella oxytoca, Pseudomonas putida* and *Bacillus licheniformis*. Interestingly, in vitro growth was maintained after long-term storage of the plant broth (up to three years at −20 °C) (Figure S2). Furthermore, plant-broth culture media of various concentrations supported sufficient collective growth of tested isolates that was comparable to, or greater than, standard culture media (nutrient agar, modified R2A, and CCM) (Figure S3).

*3.2. In Situ Recovery and Cultivability of Wheat Endophytes on Homologous Wheat Broth-Based Culture Media*

The cultivable populations of wheat endo-phyllosphere and endo-rhizosphere were assessed as CFUs that developed on surface-inoculated agar plates of various culture media. For both plant spheres, the wheat-broth culture media supported in situ development of wheat endophytes similar to that on both the standard culture media and wheat powder-based culture media (Table 1). Cultivable populations of endophytes were significantly higher (log 7.16–7.96 CFUs $g^{-1}$) in the endo-rhizosphere than those of the endo-phyllosphere (log 6.11–6.59 CFUs $g^{-1}$). Notably, micro-colonies showed a tendency to develop in endo-rhizosphere samples (40–80% of total colonies), compared to endo-phyllosphere samples (20–60%). Likewise, they were distinguishable on modified R2A and wheat powder teabag culture media.

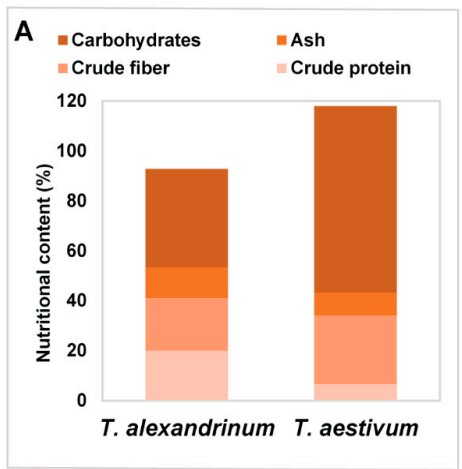

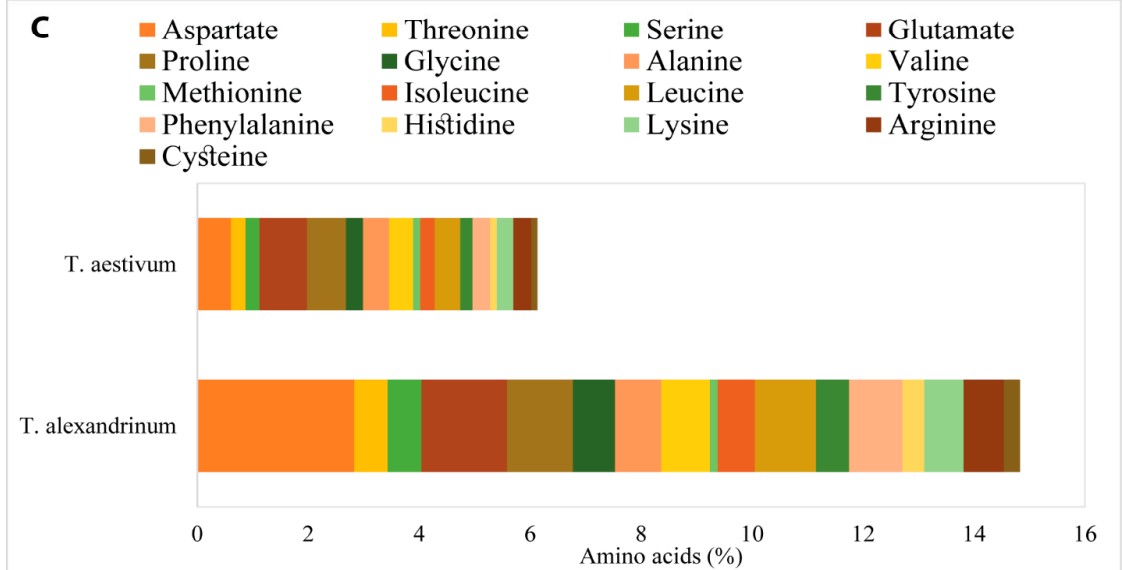

**Figure 2.** Nutritional profile of the dehydrated powders of *Trifolium alexandrinum* and *Triticum aestivum*: (**A**) Major contents of carbohydrates, proteins, fiber, and ash; (**B**) contents of macro- and micro-nutrients; and (**C**) contents of total and individual amino acids.

**Table 1.** Analysis of variance (ANOVA) analysis of log CFU counts obtained for the endo-rhizosphere and endo-phyllosphere of wheat: The two-way interaction between tested culture media and incubation time is shown.

| Culture Media | Endo-Rhizosphere (Log CFUs g⁻¹) | | | Endo-Phyllosphere (Log CFUs g⁻¹) | | |
|---|---|---|---|---|---|---|
| | Incubation at 28 °C (Days) | | | | | |
| | 2 Days | 7 Days | 14 Days | 2 Days | 7 Days | 14 Days |
| ¹/₂ modified R2A | 7.86 [abc] | 7.96 [a] | 7.94 [ab] | 6.43 [b] | 6.59 [a] | 6.46 [ab] |
| ¹/₂ CCM | 7.83 [abcd] | 7.90 [abc] | 7.76 [bcde] | 6.19 [cd] | 6.22 [cd] | 6.22 [cd] |
| WB [b] | 7.16 [f] | 7.72 [cde] | 7.62 [e] | 6.19 [cd] | 6.24 [cd] | 6.27 [c] |
| WPT [b] | 7.64 [de] | 7.92 [ab] | 7.88 [abc] | 6.11 [d] | 6.22 [cd] | 6.17 [cd] |
| HSD (*p* value ≤ 0.05) = | 0.20 | | | 0.15 | | |

Data are log means, *n* = 4; statistically significant differences are designated by different letters (*p* value ≤ 0.05, *n* =4). Modified R2A, modified Reasoner's 2A agar; CCM, N-deficient-combined carbon sources culture medium; WB, wheat broth culture medium (50 mL L⁻¹); WPT, wheat powder teabag-culture medium.

### 3.3. Cultivability of Clover Endophytes on Culture Media Prepared from Homologous (Clover) and Heterologous (Wheat) Plant Broths

In general, plant-broth culture media efficiently supported the cultivability of endophytic populations of clover in both plant compartments, the endo-phyllosphere and endo-rhizosphere (Figure S1). For the endo-rhizosphere, total CFU counts were in the range of log 8.00–9.50 CFUs g$^{-1}$, and no significant differences could be attributed to the type of culture media. In contrast, in the endo-phyllosphere, total CFU counts were much lower (log 4.00–5.50 CFUs g$^{-1}$), with highest counts developing on plant broths (clover and wheat) at the lower concentration of 25 mL broth L$^{-1}$ (Figure 3). For either plant compartment, no significant differences could be attributed to the source of plant broth culture media, based on either homologous (clover) or heterologous (wheat) broth.

For the endo-rhizosphere, incubation times of four and six days resulted in significantly higher CFUs counts, increased by more than 10–17% compared to the shorter incubation time of two days. In contrast, longer incubation did not result in significant increases in CFUs counts (<6%) of endo-phyllosphere bacteria, especially in plant broth culture media. Plant broth culture media supported relatively less slimy and confined growth of colonies in contrast to the muculent ones that developed on modified R2A agar plates. Again, longer incubation resulted in clear development of micro-colonies on all of the tested culture media. These represented 11–22% and 24–46% on broth-based culture media of tested endo-rhizosphere and endo-phyllosphere, respectively (Table S2). The corresponding percentages for plant powder teabag culture media were much higher, 50–57% and 40–56% (Table S2).

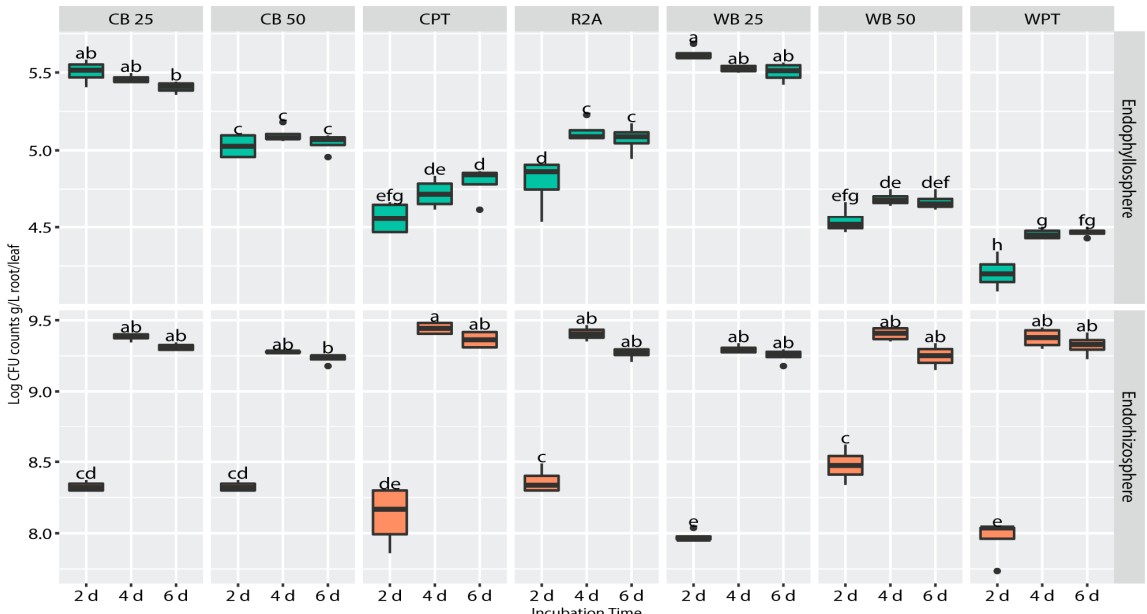

**Figure 3.** Cultivability of clover endophytes on culture media prepared from homologous (clover) and heterologous (wheat) plant broths: Two-way ANOVA analysis for cultivable endophytic bacteria of rhizosphere (orange) and phyllosphere (green) of clover plant plotted along 3 incubation times (2, 4 and 6 days); cultivations were on homologous clover broth (CB 25, 25 mL broth L$^{-1}$; CB 50, 50 mL broth L$^{-1}$) and heterologous wheat broth (WB 25, 25 mL broth L$^{-1}$; WB 50, 50 mL broth L$^{-1}$) and were compared to cultivation on teabag culture media prepared from dehydrated clover (CPT) and wheat (WPT), as well as standard chemically defined and artificial culture media ($\frac{1}{2}$ modified R2A). Statistically significant differences are indicated by different letters (*p* value ≤ 0.05, *n* = 4).

### 3.4. Diversity of Clover Endophytes Based on 16S rRNA Gene Sequences

A collection of 163 CFUs of clover endophytes were randomly selected from representative plant clover and wheat broth agar plates (25 mL broth L$^{-1}$) and modified R2A culture media. Among the

95 isolates successively sub-cultured on their corresponding culture media, 62 were successfully identified based on their good quality 16S rRNA gene sequences (Table S1). Apart from culture media, the endo-phyllosphere was represented by 30 isolates and the remainder were secured from the endo-rhizosphere (Table S3). The 62 secured isolates fell into four distinct phyla: *Firmicutes* were most prevalent (56%) followed by *Proteobacteria* (*Alphaproteobacteri*a and *Gammaproteobacteria*, 19% each), *Actinobacteria* (3%) and *Bacteroidetes* (1.6%) (Figure 4, Table S3). Whereas the modified R2A culture media isolates were confined to only the one phylum of *Firmicutes*, represented by *Bacillus* sp., *Brevibacillus* sp. and *Paenibacillus* sp., the plant broth culture media extended cultivability of endophytes to all of the four phyla (*Firmicutes*, *Proteobacteria*, *Actinobacteria* and *Bacteroidetes*), comprising nine genera (*Bacillus*, *Brevibacillus*, *Chryseobacterium*, *Cronobacter*, *Enterobacter*, *Siccibacter*, *Kosakonia*, *Rhizobium* and *Tsukamurella*).

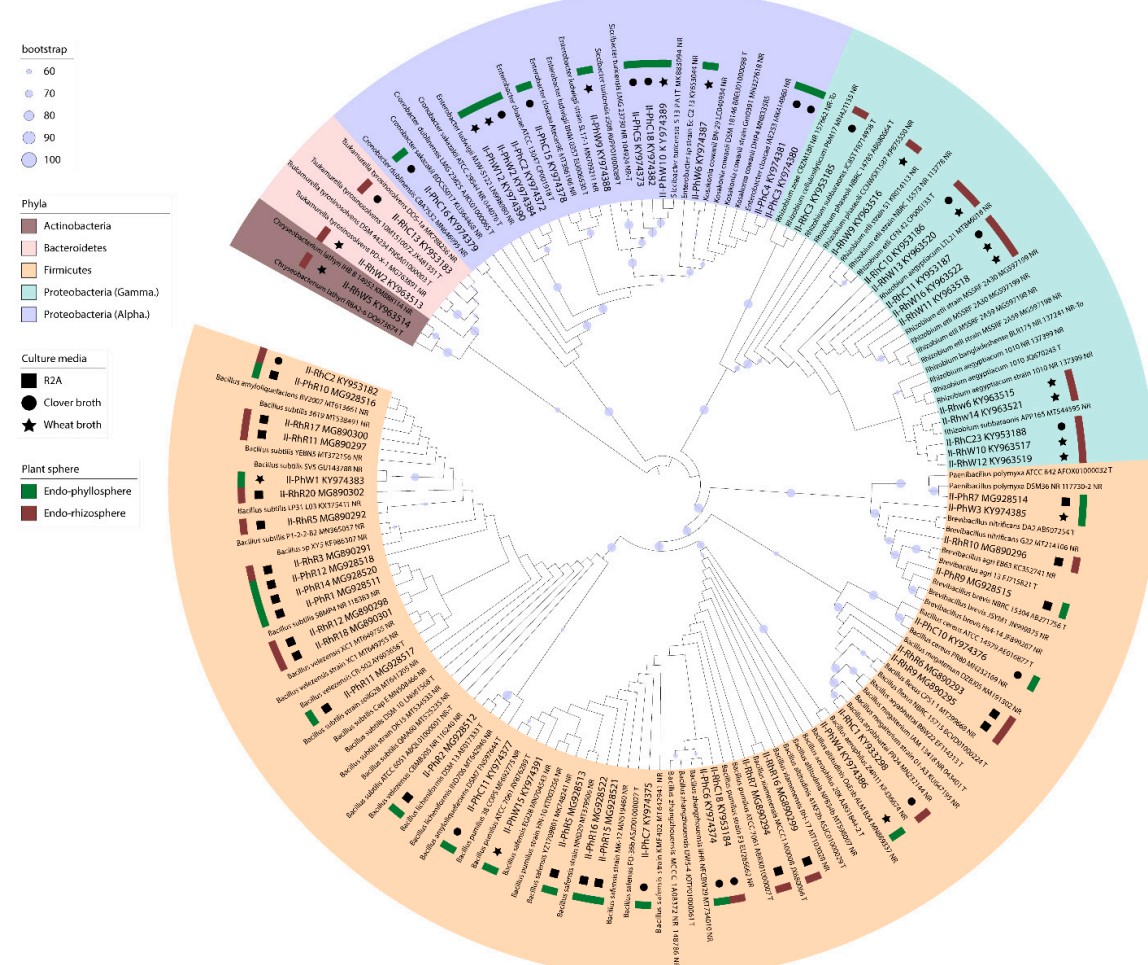

**Figure 4.** Circular phylogenetic tree based on the 16S rRNA gene sequences of 62 bacterial isolates representing endophytes of clover phyllospheres and rhizospheres. Plant broth-based culture media supported greater diversity of clover endophytes. The phylogenetic tree was constructed using Maximum Likelihood and assumes the Jukes–Cantor model of nucleotide substitution. The tree is annotated with taxonomy at phylum-level (background colour), plant sphere (coloured strip) and culture media of isolation (different symbols). Codes of isolates are indicated in bold and succeeded by their accession numbers; closest matches obtained from the GenBank database are also indicated by the organism name and accession numbers.

*Alphaproteobacteria* (*Rhizobium* sp.) prevailed in the endo-rhizosphere and were absent in the endo-phyllosphere, contrary to *Gammaproteobacteria* (*Cronobacter* sp., *Enterobacter* sp., *Siccibacter* sp.

and *Kosakonia* sp.). Furthermore, we observed an abundance of cultivable *Chryseobacterium* sp., *Cronobacter* sp., *Enterobacter* sp., *Siccibacter* sp., *Kosakonia* sp., *Rhizobium* and *Tsukamurella* sp. among clover endophytes, developed on the plant broth culture media but not on the modified R2A culture medium (Figure 5).

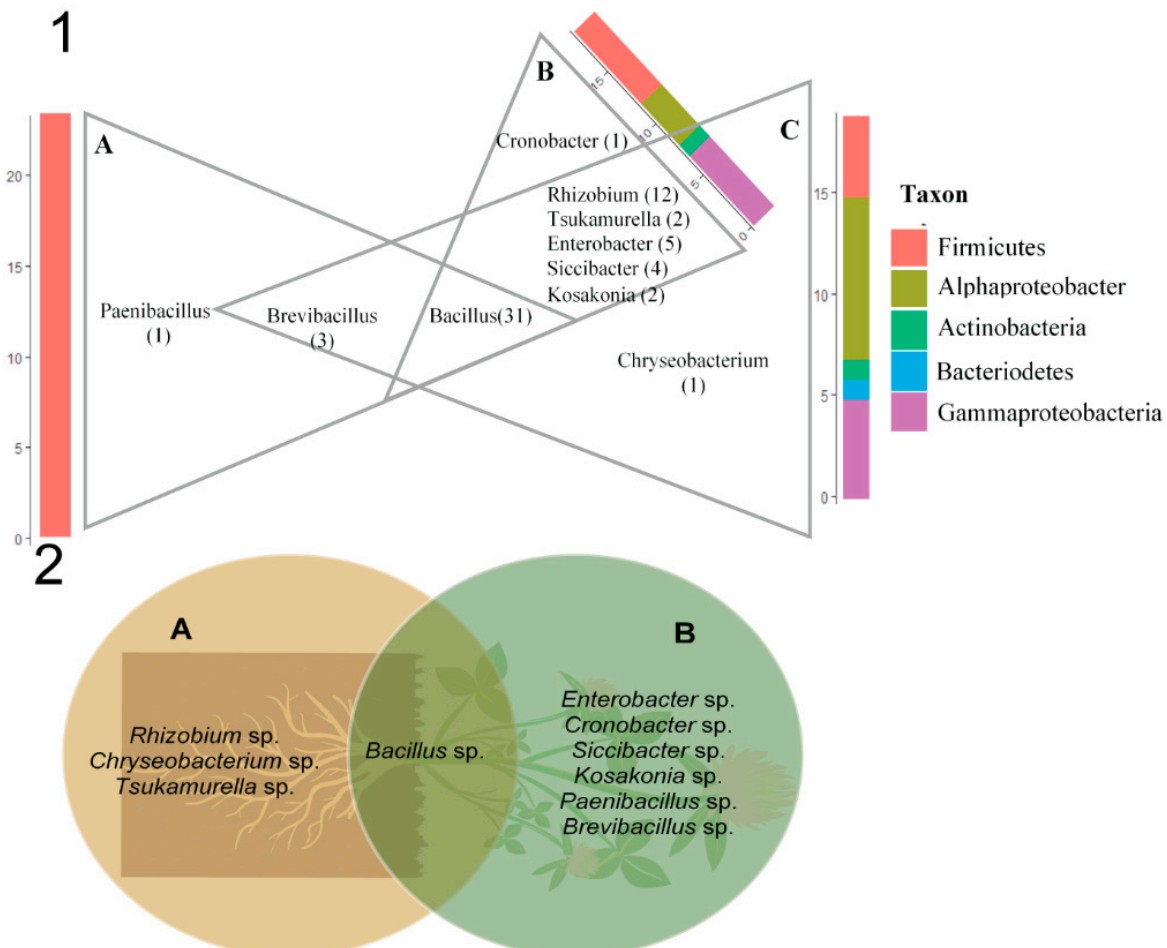

**Figure 5.** Based on 16S rRNA gene sequencing, the higher abundance of multiple genera of cultivable clover endophytes developed on plant broth-based culture media compared to modified R2A: **1**. Culture media effect: Venn diagram representing the bacterial isolates recovered by all tested culture media and their taxonomic affiliation (taxon and genera levels): (**A**) modified R2A, (**B**) clover broth culture medium, and (**C**) wheat broth culture medium. **2**. Plant compartments effect: Venn diagram representing the abundance of bacterial genera recovered in plant compartments, irrespective of culture media: (**A**) endo-rhizosphere, (**B**) endo-phyllosphere.

Among the nine bacterial genera identified, *Bacillus* commonly appeared on all tested culture media; however, *Paenibacillus* developed on modified R2A but not on plant-based culture media. Moreover, *Chryseobacterium* and *Kosakonia*.developed only on wheat broth culture media, while *Cronobacter*., *Enterobacter*., *Rhizobium*., and *Tsukamurella* appeared on both clover and wheat broth-based culture media (Figure 5).

Based on partial 16S rRNA gene sequences, and NCBI (www.ncbi.nlm.nih.gov), two isolates appeared to be potentially/presumptive novel species, when applying a threshold of approx. 98.7% 16S rRNA base similarity for species identification. Isolate II-PhR13, originated from the endo- phyllosphere and cultivated on modified R2A media, might be a novel species of *Brevibacillus*, but taxonomic affiliation relies only on a very short 242 bp fragment (96.69–97.07% identity). In contrast II-PhC15, isolated from

endo-phyllosphere on clover broth, reveals a stronger indication for a novel species, *Enterobacter* sp., based on fragment size and 98.88% identity to known closely related strains (Table S3).

### 3.5. MALDI-TOF MS Analysis of Clover Endophytes

To discriminate further between closely related isolates, the isolated CFUs were further characterized based on their protein pattern using MALDI-TOF MS. The mass spectral information generated from the most abundant proteins of an isolate was used as a fingerprint for its classification since the resolution power of this method is higher at the subspecies level compared to 16S rRNA sequences. We wanted to determine whether CFUs isolated from the two plant departments would also exhibit a similar protein pattern. Intact protein mass spectra generated were assessed for similarity using a hierarchical clustering of bacterial isolates obtained from clover- and wheat-broth culture media of endo-rhizospheres and endo-phyllospheres.

It was largely possible to group the isolates according to their respective sources, namely the endo-phyllosphere or endo-rhizosphere. Thirty-three of the clover isolates fell into two main clusters (Figure 6).

The first cluster comprised 14 isolates, the majority of which, 79%, were associated with the endo-rhizosphere. This cluster subdivided into two sub-clusters, the first occupied by *Alphaproteobacteria*, except for one isolate. The second subcluster included isolates from the endo-rhizosphere and endo-phyllosphere that belonged to *Actinobacteria* (*Tsukamurella* sp.), *Alphaproteobacteria*, *Gammaproteobacteria* and *Firmicutes* (Figure 6).

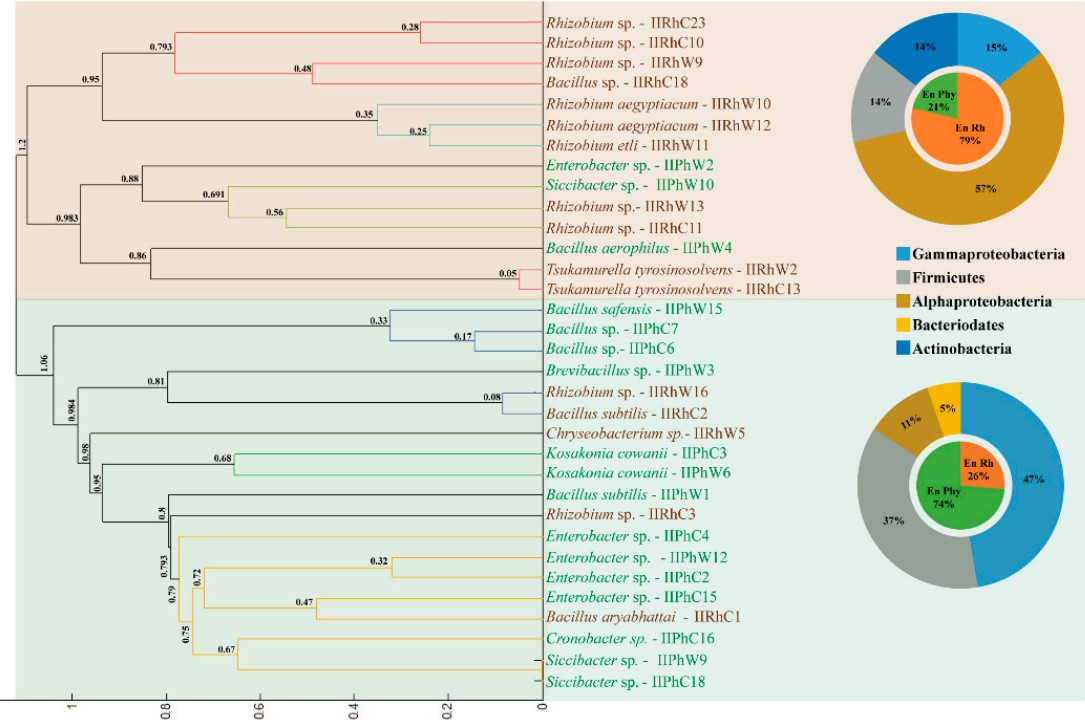

**Figure 6.** Cluster analysis of matrix-assisted laser desorption/ionization time-of-flight mass spectrometry (MALDI-TOF MS) spectra of all tested isolates recovered from clover endo-rhizosphere (in brown), and endo-phyllosphere (in green), isolated on either clover or wheat broth culture media. Potential species of isolates are indicated based on 16S rRNA gene sequencing. Distinguished are two main clusters, the brown colored background represents isolates mainly originating from the rhizosphere, and the green colored background highlights isolates mainly originating from the phyllosphere. Pie charts represent the percentages of distribution based on either plant sphere (inner pie; EnRh, endo-rhizosphere; EnPhy, endo-phyllosphere) and taxonomic level (outer ring, different taxa-related colors).

The second big cluster contained 19 isolates, the majority of which, 74%, originated from the endo-phyllosphere and classified as *Gammaproteobacteria* (*Cronobacter* sp., *Enterobacter* sp., *Siccibacter* sp. and *Kosakonia* sp.) based on 16S rRNA gene sequencing. A few isolates belonged to *Alphaproteobacteria* (*Rhizobium* sp.), *Bacteroidetes* (*Chryseobacterium* sp.) and *Firmicutes* (*Bacillus* sp.).

The clover isolates growing on their homologous clover plant-based medium were not distinct in their protein profiling compared to the isolates growing on the heterologous wheat plant broth-based medium (Figure 6).

## 4. Discussion

It is well established that throughout their life cycle, plants recruit their microorganisms from surrounding microbial repertoires of soil rhizosphere, phyllosphere, anthrosphere and spermosphere [45]. Plant microbiota may also be transferred both horizontally and/or vertically, being influenced by plant organs, and ultimately residing outside (epiphytes) or inside (endophytes) their host plants. Here, the cultivation of microbial species is indispensable and remains a major challenge to microbiologists, since pure cultures are vital for studying microbial morphology, physiology, genomes, metabolomes, and ecological impacts, as well as enabling manipulation [46,47].

Recent studies successfully developed "culturomics" as a high throughput strategy for in vitro cultivation of the human microbiota, using matrix-assisted laser desorption/ionization time of flight mass spectrometry (MALDI-TOF MS) and/or 16S rRNA amplification and sequencing to identify the developing colonies [14]. To limit fast growers and enrich rare species, various growth conditions were created and successfully tested. The extensive application of MALDI-TOF MS for rapid and high throughput identification dramatically extended the culturable human gut microbiome to levels equivalent to those of the pyrosequencing techniques.

In contrast to the major achievements of culturomics applied to gut microbiome, a major obstacle facing plant microbiome culturing is the maintained use of incompatible culture media. This is despite the unprecedented efforts and great achievements in painstakingly formulating and developing numerous culture media to date, which now exist as sets of lab media. Hence, we previously introduced plant-only-based culture media. Principally, such challenging plant media are based on juices, saps, homogenates and/or dehydrated powders of tested host plants [19,22,23,25,26,33]. Here, we further advance plant broths as a compatible milieu to foster widespread accessibility, reproducibility and reliability of plant-only-based culturing strategies.

Our results indicate that the vegetative materials of tested host plants had sufficient nutrients to prepare ample plant broths: carbohydrates (39.3–74.58%), proteins (6.65–20.0%), ash (9.20–13.30%) and multiple amino acids (0.29–2.83%). The broths supported distinctive in vitro growth of tested pure cultures representing members of the most abundant rhizobacteria, e.g., *Klebsiella oxytoca*, *Pseudomonas putida* and *Bacillus licheniformis*. Furthermore, growth indices of tested isolates on long-term stored plant broth were consistent, supporting growth at similar levels to fresh broth as well as standard culture media. These findings agree with those of Neumann and Römheld [48] and Tesfaye [49] who reported that natural plant materials contain large quantities and complex assortments of organic compounds that satisfy the nutritional needs of diverse microorganisms. These materials vary in quantity and quality with plant species, genotype, age and physiological status.

It is well established that the endophytic microbial communities in plant tissues are primarily non-specific and are selected through a combination of the available bulk soil microbial pool, plant species and environmental conditions [50,51]. In the present study we demonstrate that culture media based on plant broth of clover and wheat supported significant development of endophytic populations in both the phyllosphere and rhizosphere of tested plants. The CFUs were relatively confined and non-slimy, eliminating the coalescence of fast-growing colonies that overrun developing micro-colonies. Possibly, nutrients in the aqueous broth are present in real-time concentrations and easily accessible, allowing for the onset of prompt growth of endophytes without a lag phase.

The numbers of CFUs developing on plant-broth culture media were higher, in the majority of cases, than those recovered on modified R2A, as a reference culture media.

The tested plant broths of clover and wheat were fairly promiscuous, since they were significantly and usually able to support the general development of endophytes associated with both plants tested. The results also confirmed the "rhizosphere effect", where the rhizosphere endophytic community exceeded that of the endo-phyllosphere. Indeed, the rhizosphere microbiota extends the capacity of plants to adapt to the environment, and establishment of a particular microbiota consortium in the rhizosphere can be regarded as niche colonization.

Therefore, in situ, bacteria are equipped with necessary traits that enable them to invade, colonize and translocate in the plant's interior. Of these traits are motility, chemotaxis, production of cell-wall-degrading enzymes and lipopolysaccharide formation [51–54]. The plant transpiration stream seems to further facilitate bacterial movements inside plants [55].

The tested plant broth represents a mosaic water extract of diverse plant macromolecules, major and minor elements, as well as growth factors in the form of amino acids and other compounds of unknown composition and concentration. This creates an environment possessing a compatible vegan nutritional matrix that favors in vitro cultivability and is similar to conditions *in planta*, in the rhizosphere or phyllosphere. In contrast, the widely used standard culture media of designed and/or defined composition contain prescribed components, possibly not related, required and/or essential nutrients. They are of varying complexities, and may contain concentrated pure chemicals that might interact upon sterilization, resulting in chemical intermediates probably suppressing the growth of certain members of microbiota [56]. Probably, this explains why plant-based culture media supports better in situ bacterial recovery than designed/synthetic media.

We saw abundant growth of endophytes to the micro-colony size, particularly on diluted plant broth, which were even better detected by microscopy. This agrees with previous findings that up to 99% of the colonies that grow on diluted media develop to a micro-colony size [57]. A number of studies suggest that dilutions of conventional culture media provide nutrients with suitable concentrations of carbon and energy sources required for the growth of micro-colony-forming bacteria, and expand the range of cultivable microbial species, in particular more fastidious groups [16,58].

To look more closely at the phylogenetic diversity of cultivable endophytes, representative single-colony isolates were secured from plant shoots and roots of clover developed on plant-broth culture media as well as the chemically defined and artificial modified R2A culture media. Among the 62 successfully sequenced isolates, *Bacillus* sp. and *Rhizobium* sp. were the most common on the plant-broth culture media. In addition, the plant broth significantly extended the diversity to include the genera *Chryseobacterium*, *Cronobacter*, *Enterobacter*, *Kosakonia*, *Rhizobium*, *Siccibacter* and *Tsukamurella*. Between these isolates probably a candidate novel species could be detected. In contrast, modified R2A isolates were confined to *Paenibacillus* sp., *Brevibacillus* sp. and *Bacillus* sp. Thus, the plant broth culture media enables in vitro cultivation of a wide spectrum of plant microbiota that could potentially extend to bacterial species with probiotic functions, e.g., *Lactobacillus* sp. and *Bifidobacteria* sp. [27]. In agreement with these results was the cultivable community structure of bacteria identified from both root and inner tissues of maize as well as rice seedlings [59,60]. Our previous results [25,26,33] indicated that plant-based culture media significantly increase the cultivability of endophytic plant microbiota to include representatives of not-yet cultured genera, and less abundant and/or hard-to-culture bacterial phyla.

MADLI-TOF MS has become a well-established and rapid method for characterizing and identifying bacteria based on protein profiles [61,62]. However, the restricted database of MADI-TOF MS leads to difficulties in identifying bacteria from various environmental samples [63,64]. Preference of *Rhizobium* sp. for the endo-rhizosphere niche is well established [65], since it is common and grows well on the tested plant-broth culture media. In contrast, representatives of *Gammaproteobacteria*, *Enterobacter*, *Siccibacter*, *Kosakonia* and *Cronobacter*, were only isolated from the endo-phyllosphere. However, it is possible that isolates originating from the root might reveal protein profiles similar to

endo-phyllosphere bacteria, when growing on plant broth derived from full-grown shoot. That result further supports our hypothesis of the adaptation duration of endo-rhizospheric bacteria, when grown on shoot-based media. In general, the combination of MALDI BioTyper profiles with cross-cultivation presents the opportunity to estimate the adaptability of species to specific niches based on their functional capabilities. Nonetheless, we are aware that a profile describing the possible expression of proteins in a given environment (e.g., plant-broth culture media) may differ from the phylogenetic clustering based on one marker gene.

The effect of the rhizosphere microbiome, "rhizomicrobiome", is believed to rely upon the nature of chemical exudates, which mediate interactions via signaling molecules produced and released by both plants and microbes [49,50]. Here, it is expected that unidentified plant-derived metabolites present in the tested plant broth, but not in standard chemically defined and artificial culture media, promote exceptional in vitro growth of plant microbiota. In this respect, further investigations are required to determine the extent to which root and microbial secretions affect the microbial structure and function of the rhizosphere and, in particular, the mechanisms through which host plants assemble their rhizomicrobiomes and most importantly recruit beneficial microbial partners.

Conversely, in situ, bacteria are equipped with necessary traits that enable them to invade, colonize and translocate into the plant's interior. These traits include motility, chemotaxis, production of cell-wall-degrading enzymes and lipopolysaccharide formation [48,51–54]. Plant transpiration seems to further facilitate bacterial movement inside plants [55]. Liu [51] concluded that, to gain a more comprehensive picture of the endosphere microbiome, a combination of multi 'omics' tools are required (e.g., metagenomics, proteomics and metabolomics, as well as advancing computational data mining). Investigations optimally employing these tools may transform our understanding of bacterial endophytes and their interactions with host plants. Such information is required to breed endophyte-optimized crops, engineer endophytic microbiomes, and develop a better understanding of beneficial endophytes, in particular, how they can be attracted, maintained and adapted to enhance plant development.

## 5. Conclusions

We demonstrate that the designed, artificial and incompatible nutritional makeup of meat/bovine nutritional additives commonly used for culturing plant microbiota limited microbial diversity to only fast-growing members of *Firmicutes*. In contrast, the compatible vegan nutrition provided by a plant broth-based culture media exposed a wider scope of multiple interacting endophytes of clover, beyond just rhizobia, revealing nine genera of four phyla. These endophytes likely contribute to the nutritional and health status of the plant through direct/indirect interactions with rhizobia. Our plant-broth culturing strategy adopts the concept of "the environment selects" by creating an "in-situ-similis" vegan nutritional matrix appropriate to unmask novel microbial resources pertinent to biotechnological applications such as eco-friendly agro-biopreparations and safe for human intake probiotic preparations. The strategy is qualified to be the bedrock for future innovations in culturing methods leading towards intensive application of culturomics for exploring plant bacterial repertoires. This will substantially expand our repertoire of potential isolates for the bottom-up selection of key taxa of core microbiomes for future in situ applications aiming at sustainable crop production and mitigation of stresses due to climate change.

**Supplementary Materials:** The supplementary materials are available online at http://www.mdpi.com/1424-2818/12/11/418/s1.

**Author Contributions:** Conceptualization, N.A.H., S.R., and H.E.; Methodology, H.E., R.A.N.; K.W.; H.H.Y.; M.A.H.; M.E.-T. and M.S.S.; Software, S.P.; K.W.; M.A.H.; H.H.Y. and M.S.S.; Validation, N.A.H.; S.R.; S.P.; K.W. and M.F.; Formal Analysis, H.E., R.A.N.; H.H.Y. and M.A.H.; Investigation, H.E.; R.A.N.; M.S.S.; H.H.Y.; M.A.H. and M.R.A.; Resources, N.A.H.; S.R. and M.E.-T.; Data Curation, H.-S.A.D. and R.A.N.; Writing—Original Draft Preparation, H E.; M.F. and H.-S.A.D.; Writing—Review and Editing, H.-S.A.D.; N.A.H.; S.R.; M.A.H.; S.P. and K.W.; Visualization, N.A.H.; S.R.; M.F.; S.P. and M.A.; Supervision, N.A.H.; S.R. and M.F.; Project Administration,

N.A.H.; S.R. and M.A; Funding Acquisition, N.A.H. and S.R. All authors have read and agreed to the published version of the manuscript.

**Funding:** This research received no external funding.

**Acknowledgments:** Hegazi acknowledge the generous support of the Alexander von Humboldt Stiftung for running joint research projects with German partners at Leibniz Institute of Vegetable and Ornamental Crops (IGZ), Germany. Hegazi and Elsawey acknowledge the financial support of the German Academic Exchange Service (DAAD) for funding the Cairo University student training on "Molecular Biological Techniques for Studying Microbial Ecology" at IGZ, Germany and Cairo University, Egypt. They are grateful to all kinds of support provided by Eckhard George in his capacity as the research director of IGZ. Thanks are also extended to Birgit Wernitz and Claudia Tielesch for excellent technical support. The cartoon drawing in the graphical abstract is a warm gesture of the German artist Michael Becker. With gratitude, we acknowledge the lab support of our graduates Saad Mohamed, Ahmed T Morsi, Breksam Samir, Shereen Gamal and Reem Hamed.

**Conflicts of Interest:** The authors declare no conflict of interest.

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
