# Peer review of "Plant Broth- (Not Bovine-) Based Culture Media Provide the Most Compatible Vegan Nutrition for In Vitro Culturing and In Situ Probing of Plant Microbiota"

_diversity, doi:10.3390/d12110418_

Round 1

Reviewer 1 Report

The authors investigated culturing of plant microbiota and found that plant broth-based culture media grant the most compatible vegan nutrition for in vitro culturing and in situ probing and not bovine-based media. It is true that cultivation of bacteria started in the 19th century from meat extract but for many decades now the enrichment media have been adapted to the specific research questions and the need of the corresponding microbes. Oligotrophic and highly specialized bacteria simply cannot be enriched and cultivated with meat extract! The bias between bacteria present and cultured is known for decades. While the authors refer to the gut microbiota the bias is much more pronounced for soil, sea water or the rhizosphere. As a consequence a huge number of different media have been introduced as any modern textbook on microbiology can tell. To reduce the cultivation only to the R2A medium is extreme and in this form utterly wrong. There are a number of innovative methods, see e. g. Zervas, A., Zeng, Y., Madsen, A. M., & Hansen, L. H. (2019). Culturomics of Aerobic Photoheterotrophs in Wheat Phyllosphere Reveals Divergent Evolutionary Patterns of Photosynthetic Genes in Methylobacterium spp. Genome Biology and Evolution. 11(10):2895–2908. Therefore, the authors should be more cautious in the introduction not painting a B&W picture. These more sophisticated efforts have to be acknowledged and discussed. A convincing justification has to be given why only R2A medium was used for comparison. What about LB medium or a soil extract medium?

I am very surprised about the composition of the “R2A” medium used in this study because this is not the standard R2A medium used worldwide and sold by many companies. Why did the authors change the composition? Why did they add meat peptone?

Lines 21ff: what is here the connection to the human microbiome?

Figure 2: this figure is not easy to read because the different amino acids cannot be distinguished. There are errors in the amino acid names: therionine, aspartic, glutamic should be threonine, aspartate and glutamate.

Figure 4: only type strains should be used for comparison otherwise the chances of misidentifications are high. Please change.

Figure 6 is in this resolution not readable and has to be improved.

Lines 477ff: The statement “The widely used chemically synthetic culture media, on the other hand, contain defined chemicals of irrelevant composition of nutrients and other materials” is really a provocation because microbiologists spend more than a century optimizing the media for cultivation. Even if the authors only know R2A agar for comparison they have to be told that this medium is an excellent medium for cultivating bacteria from a broad range of habitats. But of course it is not perfect and this is why several thousand different media have been described and used for cultivation.

Lines 491ff: If you have a look at the composition of R2A you will see that this is not “chemical synthetic”.  Some basal media could be called chemical synthetic because they are defined in their chemical composition.

Author Response

//

Elsawey et al.

Manuscript ID diversity-944510

Responses to comments of reviewers

 The following table is addressing each of the comments raised by the reviewers, one by one, and in addition to changes in the manuscript:

-Responses to the reviewers are highlighted “Green”

-English editing from bioXpress Editorial, Revision, Writing and Translation Services, marked as Avril Arthur

With best regards, and appreciate very much your kind cooperation

nabil

Reviewer 1

Comments

Responses (line numbers refer to the file “diversity-944510 (1)-including English edits and revised marks.doc”*

To reduce the cultivation only to the R2A medium is extreme and in this form utterly wrong. There are a number of innovative methods, see e. g. Zervas, A., Zeng, Y., Madsen, A. M., & Hansen, L. H. (2019). Culturomics of Aerobic Photoheterotrophs in Wheat Phyllosphere Reveals Divergent Evolutionary Patterns of Photosynthetic Genes in Methylobacterium spp. Genome Biology and Evolution. 11(10):2895–2908. Therefore, the authors should be more cautious in the introduction not painting a B&W picture. These more sophisticated efforts have to be acknowledged and discussed. A convincing justification has to be given why only R2A medium was used for comparison. What about LB medium or a soil extract medium?

The article, as appeared in the title, is not primarily aiming at under estimating the great achievement of R2A as a very important culture medium with great achievements in the field of culturability. BUT underline the compatibility and feasibility of the use of the natural plant media in the form of plant broth to further explore the diversity of the plant microbiota.

This specific point was highlighted/acknowledged in the introduction Lines 152-154) and discussions (lines 1073-1078)

I am very surprised about the composition of the “R2A” medium used in this study because this is not the standard R2A medium used worldwide and sold by many companies. Why did the authors change the composition? Why did they add meat peptone?

R2A medium used is basically of Reasoner and Geldreich (Ref. 34) with a slight modification to further enrich the medium (https://assets.fishersci.com/TFS-Assets/LSG/manuals/IFU112543.pdf).

The text is adjusted and highlighted accordingly

(Material and Methods: lines 357-362)

Lines 21ff: what is here the connection to the human microbiome?

It is just to highlight progress of culturability of the plant microbiome compared to other environmental microbiomes. The sentence  (lines 25-27) was readjusted as follows:

“Despite recent advances of high-throughput methods, culturability is lagging behind other environmental microbiomes, notably human microbiome.”

Figure 2: this figure is not easy to read because the different amino acids cannot be distinguished. There are errors in the amino acid names: therionine, aspartic, glutamic should be threonine, aspartate and glutamate

Agree, the figure is very compact; however, and after improvement we request the understanding of the reviewer as it is pooling the nutritional profiles, macro- micro-molecules and major-minor-elements, of tested plants in a concise not wordy and lengthy form.

The figure is adjusted for better resolution and colours identification, and the names of amino acids are corrected as indicated.

Figure 4: only type strains should be used for comparison otherwise the chances of misidentifications are high. Please change.

Thanks for your recommendation. We reassigned the sequences including all respective type strains and corrected all related parts in the figures 5 and 6 and in the text as well

Figure 6 is in this resolution not readable and has to be improved

We removed the old Fig. 6 and the corresponding text

Lines 477ff: The statement “The widely used chemically synthetic culture media, on the other hand, contain defined chemicals of irrelevant composition of nutrients and other materials” is really a provocation because microbiologists spend more than a century optimizing the media for cultivation. Even if the authors only know R2A agar for comparison they have to be told that this medium is an excellent medium for cultivating bacteria from a broad range of habitats. But of course it is not perfect and this is why several thousand different media have been described and used for cultivation.

We do apologize for the misunderstanding.

We all acknowledge and appreciate the development of myriad formulas of artificial culture media and the unprecedented efforts of fellow researchers along the years to improve culturability of microorganisms in various environments. Over a century of successive efforts, now it exists a set of lab media that have been painstakingly and developed to date.

In our previous publications we used several formulas of such culture media. In this publication we particularly used R2A for comparisons, being reported as an excellent medium for cultivation of bacteria present in various environments. This is primarily to compare the performance of natural plant-only-based natural culture media to one of the best and well-known standard/artificial/designed culture media.

Accordingly, the related passage is re-edited to take into consideration the reviewer points of view (Discussion, lines 1073-1078):

In contrast, the widely used standard culture media of designed and defined composition, on the other hand, contain prescribed substrates, possibly not very much related, required and/or demanded nutrients..”

Lines 491ff: If you have a look at the composition of R2A you will see that this is not “chemical synthetic”.  Some basal media could be called chemical synthetic because they are defined in their chemical composition

Agree, this is a difficult issue . However, throughout the manuscript “chemically-synthetic” is replaced by “Standard chemically-defined and artificial culture media”. let me hope that this term will be accepted by the reviewer

Reviewer 2 Report

The paper by Elsawey et al., explores the possibility to use Plant Broth based media to isolate plant endophytes. They compare the home made media to synthetic and Bovine based ones.

The results are interesting and show the possibility to use these plant based broth media for the characterization of the plant endophytes. The results are interesting but the writing should be largely improved and some experimental details are missing.

First the plant growth conditions, including substrate, origin of soil, and presence of nodules (on clover roots) are missing in the description. Are the plant grown in green-house or in field?. If roots of the legume plants included nodules it is normal to observed higher content of rhizobia in the endophyte population.

For the writing the authors should avoid using words like excellent, exceptionally… like in paragraph 3.1. This part should be written in a neutral way giving only facts. I propose that an English native speaking person correct the whole text of the paper.

The authors should check the text police through the text. It is not always the same like in the introduction part or in 2.6.

In 3.3 the authors claim that the colonies are less slimy on the plant broth media. This is not really what is observed on Figure S3.

The number of colonies identified from rhizosphere does not really change on the different media, but was significantly higher on plant broth (25ml) for the endophyllosphere. This higher recovery at the lowest broth concentration raises the question of testing even lower broth concentrations for this compartment.

The characterization (16S) of the isolated bacteria on the different media shows that the diversity was higher on plant broth media. This is the interesting point of the paper.  

The quality of Figure 6 is very poor.

I did not understand the meaning of the section 3.5 (Maldi-TOF MS). This section should be better written/explain.

Line 478: irrelevant composition. This is the kind of sentence that should be removed from the text.

In summary I think the paper should be better presented and the writing improved.

Author Response

//

Comments

Responses

First the plant growth conditions, including substrate, origin of soil, and presence of nodules (on clover roots) are missing in the description. Are the plant grown in green-house or in field?. If roots of the legume plants included nodules it is normal to observed higher content of rhizobia in the endophyte population.

The sampled clover plants were grown in the Giza open fields at the experimental station of Faculty of Agriculture, Cairo university.

Yes nodules were present on the sampled and processed roots; which is confirmed by the isolation of more than 12 isolates of Rhizobium sp. throughout the study. Rhizobia were among the most common isolates,

The plant broth successfully recovered rhizobia. Among the sequenced isolates, Bacillus sp. and Rhizobium sp. were the most common on the tested plant-broth culture media (Discussion, lines 627-628).

The related text was adjusted accordingly (Materials and Methods, lines 133-135), and photo of root nodules is included in figure S3

For the writing the authors should avoid using words like excellent, exceptionally… like in paragraph 3.1. This part should be written in a neutral way giving only facts. I propose that an English native speaking person correct the whole text of the paper.

Agree, the paragraph is adjusted accordingly

English language of the manuscript was checked by an English language expert/service: bioXpress Editorial, Revision, Writing and Translation Services

In 3.3 the authors claim that the colonies are less slimy on the plant broth media. This is not really what is observed on Figure S3.

Agree, it is very relative

Slimy colonies were resolved on all tested culture media. However, their percentages were relatively higher on standard R2A, to the extent that colonies diffuse together in running slimy flocks (demonstrated in plates A of Fig. S3).

The text was adjusted and highlighted accordingly (Discussion, lines -1038588-589

The number of colonies identified from rhizosphere does not really change on the different media, but was significantly higher on plant broth (25ml) for the endophyllosphere. This higher recovery at the lowest broth concentration raises the question of testing even lower broth concentrations for this compartment.

Yes, further experiments, on cross-cultivation of endophytes on homologous and heterologous plant-based culture media, included concentrations of plant broth down to 5ml. Very interesting results are obtained and currently under preparation for publication.

The characterization (16S) of the isolated bacteria on the different media shows that the diversity was higher on plant broth media. This is the interesting point of the paper.  

Thanks

The quality of Figure 6 is very poor.

After improving the resolution of the figure, we prefer not to include it in the manuscript.  As the figure is solely based on presumptive taxonomic position of partial sequences of 16S rRNA sequences

I did not understand the meaning of the section 3.5 (Maldi-TOF MS). This section should be better written/explain.

The section of (Maldi-TOF MS) is edited and readjusted for better explanation, in results (lines 492-504) and discussion (641-655)

Line 478: irrelevant composition. This is the kind of sentence that should be removed from the text.

Thanks for your comment, text adjusted accordingly

(Discussions, lines 610-615)

Round 2

Reviewer 1 Report

The authors did a good job in the revision and addressed and corrected most issues. However, I am not satisfied with their explanation for the odd composition of their R2A medium used in their study. If they check R2A medium from main suppliers, e. g. Merck-Millipore, Oxoid, or the medium composition given by leading strain collections, e. g. ATCC, DSMZ, they will always find R2A medium devoid of meat peptone. As this point is very crucial both in the working hypothesis of this study and its results we require here an excellent and very convincing explanation why this deviating R2A composition has been used for comparison.

Author Response

//

  1. Thanks for accepting our responses to the raised comments and approving the adjustment of the main text.
  2. Agree that the composition of the used R2A medium is not of the original one of major suppliers, Merck-Millipore, Oxoid, or given by leading strain collections, e. g. ATCC, DSMZ. Here, we draw your kind attention to the following:
  • The main objective of the study was comparing the plant-broth culture media to one of the commonly used bovine-based recommended culture media, e. R2A as an example.
  • Because of the unavailability of the dehydrated product of R2A medium as well as the component proteose peptone in our laboratory and at local lab suppliers in Egypt for many reasons including export restrictions. With search, we get across the modified R2A formula (https://assets.fishersci.com/TFS-Assets/LSG/manuals/IFU112543.pdf), that all of its ingredients are available to prepare the medium. As both R2A and modified R2A are considered bovine-based culture media (Table below), we commonly use such modified R2A medium in our lab and already included in several of graduation projects, MSc theses and recent publications (Rahma A. Nemr, et al. 2020 .In situ similis” Culturing of Plant Microbiota: A Novel Simulated Environmental Method Based on Plant Leaf Blades as Nutritional Pads  Microbiol., 07 April 2020, (11) 1-15. https://doi.org/10.3389/fmicb.2020.00454). 
  • To comply with the above mentioned information, it was imperative to watch that carefully in the title of the manuscript and in section materials and methods:
    • The  title of the manuscript Plant broth- (not bovine-) based culture media provide the most compatible vegan nutrition for in vitro culturing and in situ probing of plant microbiota”,  focused on bovine-based culture media in general not R2A in particular
    • In the section Materials and methods we stated/specified: Standard chemically-defined and artificial culture media R2A agar, with a slight modification that contains (g L−1): casein hydrolysate, 0.5; dextrose, 0.5; soluble starch, 0.5; yeast extract, 0.5; dipotassium phosphate, 0.3; sodium pyruvate, 0.3; casein peptone, 0.25; meat peptone, 0.25; magnesium sulfate, 0.024. Agar was added (2% w/v) and pH adjusted to 7.0 ± 0.2 [34; https://assets.fishersci.com/TFS-Assets/LSG/manuals/IFU112543.pdf].
  • To be more specific, we further adjusted the main text to all over replace R2A with modified R2A (highlighted in green). This is to assert that the presented results are still valid in comprehensively comparing the plant-broth with common bovine-based culture media.

R2A media (g L-1)

Reasoner and Geldreich (1985)

Modified R2A  (g L-1) ,         https://assets.fishersci.com/TFS-Assets/LSG/manuals/IFU112543.pdf

Comments

Casamino acids            0.5

Casein hydrolysate                 0.5

Roughly, both are similar, but there are processing differences- the real difference is filtering

Proteose peptone         0.5

Casein peptone                     0.25

Meat peptone                       0.25

All of these bacteriological pre-digested peptones  are of bovine source, not plant or yeast; being organic nitrogen source containing the short chained peptides & broad spectrum amino acids required for general microbial growth

Yeast extract                0.5

Yeast extract                        0.5

Dextrose,                         0.5

Dextrose                                0.5

Soluble starch             0.5

Soluble starch                     0.5

Dipotassium phosphate   0.3

Dipotassium phosphate      0.3

Magnesium sulfate     0.05

Magnesium sulfate               0.024

Sodium pyruvate         0.3

Sodium pyruvate                   0.3

Agar       15.0

Final pH 7 ± 0.2 @ 25 °C

Agar       15.0

Final pH 7 ± 0.2 @ 25 °C                                  

Reviewer 2 Report

Thank you for the different improvements. This makes now the manuscript acceptable for publication.

Author Response

Many thanks for accepting our responses to the comments raised, and to the adjusted manuscript

Round 3

Reviewer 1 Report

I agree to the solution found by the authors to label their version of R2A as "modified R2A".